# A portable lateral flow distance-based paper sensor for drinking water hardness test

**Yulin Liu**[1,2], **Longzhan Dong**[2], **Wenli Wu**[1], **Jiantao Ping**[1], **Jingbo Chen**[2]*, **Qiongzheng Hu**[1]*

1 Shandong Analysis and Test Center, Qilu University of Technology (Shandong Academy of Sciences), Jinan, China, 2 Department of General Surgery, The First Affiliated Hospital of Shandong First Medical University, Jinan, China

* huqz@qlu.edu.cn (QH); qychenjingbo@163.com (JC)

## Abstract

Hardness is one of the basic parameters of water, and a high-level hardness of drinking water may be harmful to human health. Thus, it is very important to monitor drinking water hardness. In this work, a portable lateral flow distance-based paper sensor for the semi-quantitative detection of drinking water hardness is demonstrated. In the presence of $Ca^{2+}/Mg^{2+}$, the hydrogel can be formed via the chelation between sodium alginate and $Ca^{2+}/Mg^{2+}$, inducing a phase separation process. The viscosity change of the sodium alginate solution is directly related to the $Ca^{2+}/Mg^{2+}$ concentration and can be determined by the water lateral flow distance on test strips. The sensor successfully realizes the quantification of $Ca^{2+}$ and $Mg^{2+}$ in the range of 0–10 mmol $L^{-1}$ and 4–20 mmol $L^{-1}$, respectively. The recoveries are found varied from 95% to 108.9%. The water hardness is acceptable for drinking if the $Cr$ values lies in the range of 0.259 to 0.419, and it is high with the $Cr$ value above 0.595. Remarkably, the performance of the sensor is comparable with the commercial kit for real water samples, which avoids the subjective judgment. Overall, this method provides a portable approach for semi-quantitative detection of drinking water hardness with the merits of convenience and low cost, which shows great potential for the potential application.

## Introduction

Water is an indispensable natural resource for human beings. Drinking water quality is vital to human health. Water hardness is one of the basic parameters for evaluating water quality, and a high-level drinking water hardness may cause diseases [1, 2]. Water hardness is generally referred to the sum content of calcium, magnesium, iron, aluminum, zinc and other ions contained in water, and usually calculated by $Ca^{2+}$ and $Mg^{2+}$ contents considering the lower concentrations of other ions, which is also called as $Ca^{2+}$ hardness. Although the $Ca^{2+}$ concentration in drinking water is not strictly defined, the World Health Organization recommends that calcium ion level in drinking water should be no more than 5 mmol $L^{-1}$ [3, 4]. Therefore, it is very important to detect $Ca^{2+}$ hardness in drinking water samples.

Until now, several methods for monitoring $Ca^{2+}$ have been reported despite its difficulty to be differentiated from other interfering ions. Among them, atomic absorption spectrometry

**Funding:** Natural Science Foundation of Shandong Province (ZR2021QH106, ZR2020QB153, ZR2022YQ123); the Taishan Scholars Program (tsqn201812088); the Education and Industry Integration Pilot Project of Qilu University of Technology (2023PY058, 2022PY036) The funders had no role in study design, data collection and analysis, decision to publish, or preparation of the manuscript.

**Competing interests:** The authors have declared that no competing interests exist.

and complexometric titration are the classic methods used for calcium quantification [5, 6]. However, they often suffer from the disadvantages such as complex procedures, bulky sample requirement, complicated instrumentation, and trained operators. For instance, the commercial colorimetric kit has already been accessible, which is developed based on the titration method. However, a large amount sample of 10 mL was generally required. Meanwhile, the color change of titration terminal is subjectively judged by operators, which may inevitably lead to systematic errors. Additionally, the fluorescence method is available for the simplified detection process with high sensitivity [7–9], but it still requires the usage of large-scale instrument and trained operators. Ionophore-based ion-selective optode also have been used for the colorimetric detection of $Ca^{2+}$ [10, 11]. Nevertheless, the results are greatly influenced by the pH of the samples, which hampers their broad application. Thus, it is greatly demanded to construct a portable sensor for monitoring $Ca^{2+}$ in water.

Paper-based detection methods have become appealing in recent years, with the merits of convenient operation, fast response, and easy modification. They are extensively applied in clinical diagnostics, food quality management, and environmental monitoring [12–15]. In particular, the distance-based paper sensor can quantify the analyte by measuring the change of water flow distance, which has shown great application prospect as point-of-care test, because of its advantages of simple portability, visualization, easy quantification, and short analysis time [16–18]. The pH indicator papers can be employed as test strips because they are cheap and can clearly show the water flow marks. In addition, analyte-responsive hydrogels are series of polymers with three dimensional network structures. Various hydrogels have been designed and developed for biosensing of metal ions [19, 20], nucleic acids, proteins [21], and microorganisms [22, 23]. With the specific experimental design, hydrogels can respond to analytes and induce physical or chemical changes that generate subsequent readable signals. Among these methods, the gel-sol transition triggered by external stimuli is a most commonly used detection principle [23–25]. Therefore, it possesses great potential for the exploration of the distance-based lateral flow sensor using stimuli-responsive polymers for the evaluation of drinking water hardness.

Herein, a portable lateral flow sensor with the distance readout signal for the semi-quantitative determination of drinking water hardness on the paper strip was developed (Scheme 1). In the presence of $Ca^{2+}/Mg^{2+}$, the sodium alginate (Alg) hydrogel network with "egg-box" structure can be formed during the phase separation process. Correspondingly, the viscosity of the Alg solution drops sharply due to reduction of the Alg concentration. Thus, the concentration of $Ca^{2+}/Mg^{2+}$ can be determined via the measurement of water lateral flow distance on paper strips of the residual Alg solution. This method offers a simple and convenient method for the water $Ca^{2+}$ hardness evaluation using a small amount of samples with satisfacoty accuracy, which also avoids the subjective color endpoint judgment.

## Experimental

### Materials

Sodium alginate (CP, viscosity 200 ± 20 mPa·s), $(NH_4)_2S_2O_8$ (AR, 98.1%), and $NaHSO_3$ (AR, 99.99%) were obtained from Macklin. Hydrochloric acid (AR) for pH adjustment was obtained from Sinopharm Group. $NaH_2PO_4$ (AR, 99.0%) and $Na_2HPO_4$ (AR, 99%) were purchased by Aladdin Technology Co., Ltd., China. Water hardness test kit was bought from Lohand Biological. Co., Ltd. The pH strips with the dimension of 60 mm × 5 mm (length × width) were used in this investigation. The PVC plates were cleaned by ethanol thoroughly and dried before use. Tris-HCl buffer (pH = 7.4) was offered from Sangon Biotech Co., Ltd., China.

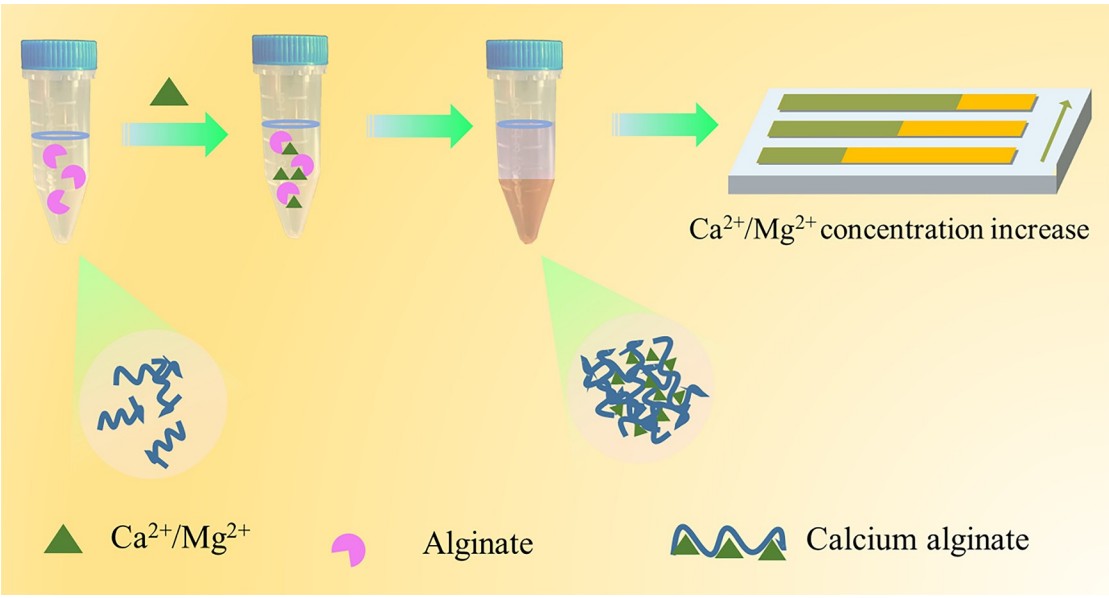

**Scheme 1. The principle of the portable distance readout paper sensor for evaluating drinking water hardness.**

### Optimization of the sodium alginate concentration

Tris-HCl buffer (100 mmol $L^{-1}$, pH = 7.4) was used in this study. The Alg solutions with different concentrations from 0.1 wt% to 0.5 wt% were firstly prepared at 25°C. The solutions of $CaCl_2$ with various concentrations of 0, 2, 4, 10, 20 mmol $L^{-1}$ were then obtained. According to our previous study, the mixture of $CaCl_2$ and Alg solution was incubated at 25°C after vortex for 30s [26]. Then, 30 μL of the supernatant solution obtained by centrifuging for 1 min was transferred onto the left side of the test strip. After waiting for 2 min, the images of the paper sensor were captured by smartphone and analyzed by Adobe Photoshop software. All of the experiments were conducted at least three times to obtain the standard deviations.

### Determination of the water hardness using the commercial test kit

Firstly, 10 mL of the test solution was accurately pipetted into the cleaned conical flask, followed by the addition of a package of total hardness reagent I. If the solution color is pure blue after reagent I dissolved, the hardness value of the water sample is 0 mg/L. If the solution exhibits purple red, hardness reagent II aqueous solution was then vertically added with the continuous shaking of the conical flask. The dropping speed should be controlled as 1 drop every 3 s until the solution changed from purple red to pure blue. Then, the number of drops consumed (N) was recorded and hardness (mg $L^{-1}$, calculated as $CaCO_3$, 1 mg $L^{-1}$ = 0.01 mmol $L^{-1}$) was calculated by the following equation, the detection range of this kit is 30–600 mg $L^{-1}$:

$$Hardness = N \times 30$$

### Data analysis

The pH indicator papers were photographed with a smartphone. Then, the pixel values of areas including water marked and whole test strips were obtained by Adobe Photoshop and recorded as $P_{mark}$ and $P_{total}$, respectively. Finally, the ratio of $P_{mark}$ and $P_{total}$ was calculated as

the water trace coverage ratio ($Cr$) as follows:

$$Cr = P_{mark}/P_{total}$$

## Results and discussion

### Feasibility of the paper sensor for the evaluation of Ca²⁺/Mg²⁺

The responses of the paper sensor in different conditions were recorded. As shown in Fig 1, the flow distance of the $Ca^{2+}$ solution was obviously longer than that of the 0.2 wt% Alg solution. The $Cr$ values were 0.8 and 0.5, respectively. A similar phenomenon also occurred for the $Mg^{2+}$ solution (S1 Fig). The difference was mainly ascribed to the viscosity discrepancy caused by Alg itself. Upon the Alg solution was incubation with the 10 mmol L⁻¹ $Ca^{2+}$ solution, the $Cr$ value increased to 0.65 with the longer water flow distance as shown in Fig 1A and 1B. The solution viscosity was increased obviously in the existence of Alginate compared with that of $Ca^{2+}$ and $Mg^{2+}$ existing alone in Fig 1C. These results are consistent with the proposed principle above. The viscosity decrease of the Alg solution was caused by the consumption of Alg via cross-linking with $Ca^{2+}/Mg^{2+}$ during the formation of the hydrogel [26–29]. Overall, this method provides an effective means to monitor $Ca^{2+}/Mg^{2+}$, which is candidate for the drinking water hardness determination.

### Optimization of condition for the paper sensor

The performance of Alg solutions at different concentrations on pH test strips was studied. As shown in Fig 2A and S2 Fig, with the concentration of Alg solution elevating from 0.1 wt% to 0.5 wt%, the water flow distance descended obviously due to the increasing viscosity. The

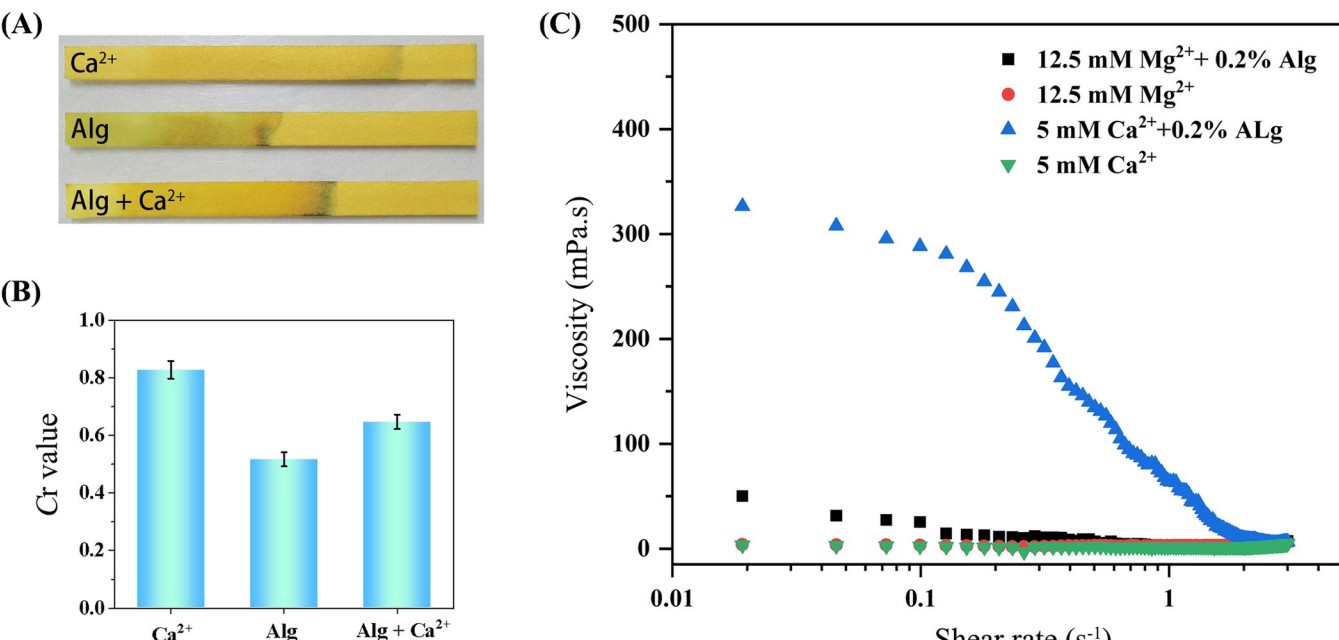

**Fig 1. Feasibility of the paper sensor for the evaluation of Ca²⁺/Mg²⁺.** (A) The photopgraphs and (B) the $Cr$ values of the paper sensor in the $Ca^{2+}$ solution, the Alg solution, and the mixture of Alg and $Ca^{2+}$, respectively. The concentrations of $Ca^{2+}$ and Alg are 10 mmol L⁻¹ and 0.2 wt%, respectively. (C) The viscosities of $Ca^{2+}$, $Mg^{2+}$, Alg/ $Ca^{2+}$ and Alg/Mg²⁺ solution, respectively.

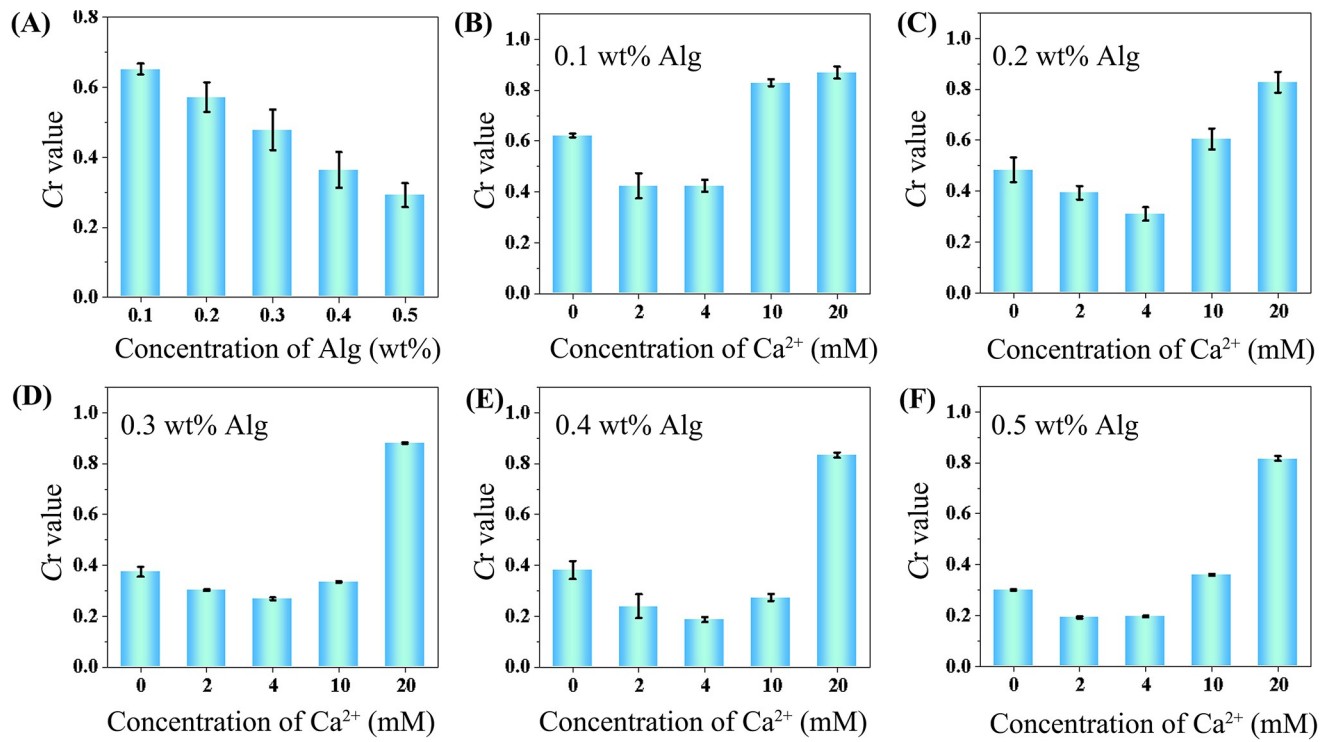

**Fig 2. Optimization of conditions for the paper sensor.** (A) The $Cr$ values of the paper sensor responses towards the Alg solutions with different concentrations. (B)-(F) The responses of the paper sensor towards different concentrations of Alg with different concentrations of $Ca^{2+}$ (0, 2, 4, 10 and 20 mmol $L^{-1}$).

concentration of $Ca^{2+}$ from 0 to 20 mmol $L^{-1}$ was evaluated at a fixed Alg concentration of 0.1 wt%. Surprisingly, the water flow distance becomes shorter with the concentration increase of Alg from 0–4 mmol $L^{-1}$, while further distance increases with the Alg concentration from 4 to 20 mmol $L^{-1}$ (Fig 2B and S3 Fig). Moreover, the similar variation tendency was found for the solution and the mixture of Alg and $Mg^{2+}$ (S1 Fig), respectively. The concentrations of $Ca^{2+}$ and Alg were 10 mmol $L^{-1}$ and 0.2 wt%, respectively. Moreover, the similar variation tendency was observed with different concentrations of Alg (Fig 2C–2F). The results are reasonable considering the crosslinking process. At the initial stage, the addition of $Ca^{2+}$ into Alg solution can cause the rapid formation of the hydrogel based on the chelation between them, leading to the increased viscosity. While the continued increase of the $Ca^{2+}$ concentration contributed to increase of the cross-linking degree of the hydrogel, which was separated from the aqueous phase. As only the remainly Alg was in the aqueous phase, the viscosity is decreased with further increase of the $Ca^{2+}$ concentration. Thus, the viscosity change can be coupled to the concentration change of $Ca^{2+}$. As concluded from Fig 2, the fixed 0.2 wt% Alg solution was chosen for following experiments, because it can clearly distinguish different concentrations of $Ca^{2+}$.

## Detection of $Ca^{2+}$ and $Mg^{2+}$

Inspired by the excellent performance of the paper sensor above, the study investigating determination of the concentration of $Ca^{2+}$ and $Mg^{2+}$ was also conducted. Based on the results in Fig 2, the $Cr$ value dropped at the $Ca^{2+}$ concentration from 0–4 mmol $L^{-1}$ and increased at the $Ca^{2+}$ concentration from 4–20 mmol $L^{-1}$, which makes it challenging for the direct quantitative detection of $Ca^{2+}$. Thus, 4 mmol $L^{-1}$ $Ca^{2+}$ was initially added directly into the test sample to

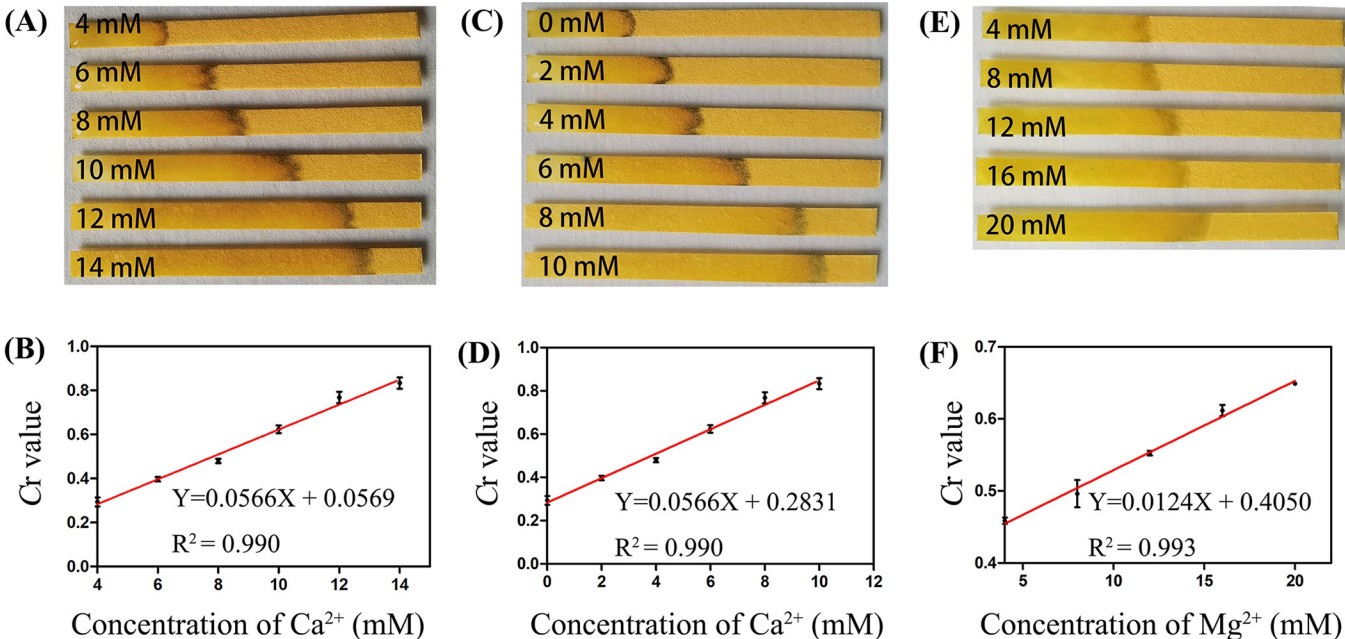

**Fig 3. Detection of Ca$^{2+}$ and Mg$^{2+}$.** (A) The images of the sensors and (B) the linear plot between the $Cr$ and Ca$^{2+}$ concentration from 4–14 mmol L$^{-1}$. (C) The images of the sensors and (D) the linear plot between the $Cr$ and Ca$^{2+}$ concentration from 0–10 mmol L$^{-1}$. (E) The images of the sensors and (F) the linear plot between the $Cr$ and Mg$^{2+}$ concentration from 4–20 mmol L$^{-1}$.

address this issue. In this way, the concentration of Ca$^{2+}$ can be quantitatively determined. As shown in Fig 3A, the water flow distance increased complying with the progressive concentration increase in the range of 4–14 mmol L$^{-1}$. Thus, the function between $Cr$ value and the Ca$^{2+}$ concentration from 4–14 mmol L$^{-1}$ was plotted as Fig 3B. A satisfactory linear relationship was obtained with R$^2$ as 0.990. With the initially added 4 mmol L$^{-1}$ Ca$^{2+}$ case, the function exhibited the same trendency in the range of 0–10 mmol L$^{-1}$ (Fig 3C and 3D). Besides, the responses of the paper sensor to Mg$^{2+}$ were also studied and the pictures were showed in S3 Fig. The $Cr$ value exhibited decreasing trendy accompanied by the Mg$^{2+}$ concentration rising from 0 to 4 mmol L$^{-1}$. Following, the $Cr$ values increased from 4–16 mmol L$^{-1}$ and reached a plateau until the concentration of Mg$^{2+}$ increased to 20 mmol L$^{-1}$ (Fig 3E and 3F). A linear relationship was also plotted with R$^2$ calculated to be 0.993. The results showed the potential of the paper sensor for Ca$^{2+}$ and Mg$^{2+}$ hardness detection in water.

## The selectivity of the paper sensor

The Ca$^{2+}$ of 10 mmol L$^{-1}$ was chosen in this section. The water flow distance remained almost same with different pH values from 5–9 (Fig 4A), which indicated the pH stability of the paper sensor. Subsequently, the influence of ionic strength of the solution was also considered (Fig 4B). The actual drinking water samples rarely contains heavy metal ions. Therefore, common anions were selected for selectivity detection. The results show that ignorable difference was observed at a fixed Ca$^{2+}$ concentration of 10 mmol L$^{-1}$ in the presence of NaCl solution (0–200 mmol L$^{-1}$). The corresponding photograph is shown in S4 Fig. Finally, the potential interfering ions in real samples including NaCl, KCl, NaH$_2$PO$_4$, NaH$_2$PO$_4$, (NH$_4$)$_2$S$_2$O$_8$, Na$_2$S$_2$O$_3$, NaHSO$_3$ and NaHCO$_3$ were mixed with Ca$^{2+}$. As shown in Fig 4C and 4D, the water flow distances exhibited no obvious alternation. All of these results suggest the satisfactory **selectivity** of the paper sensor.

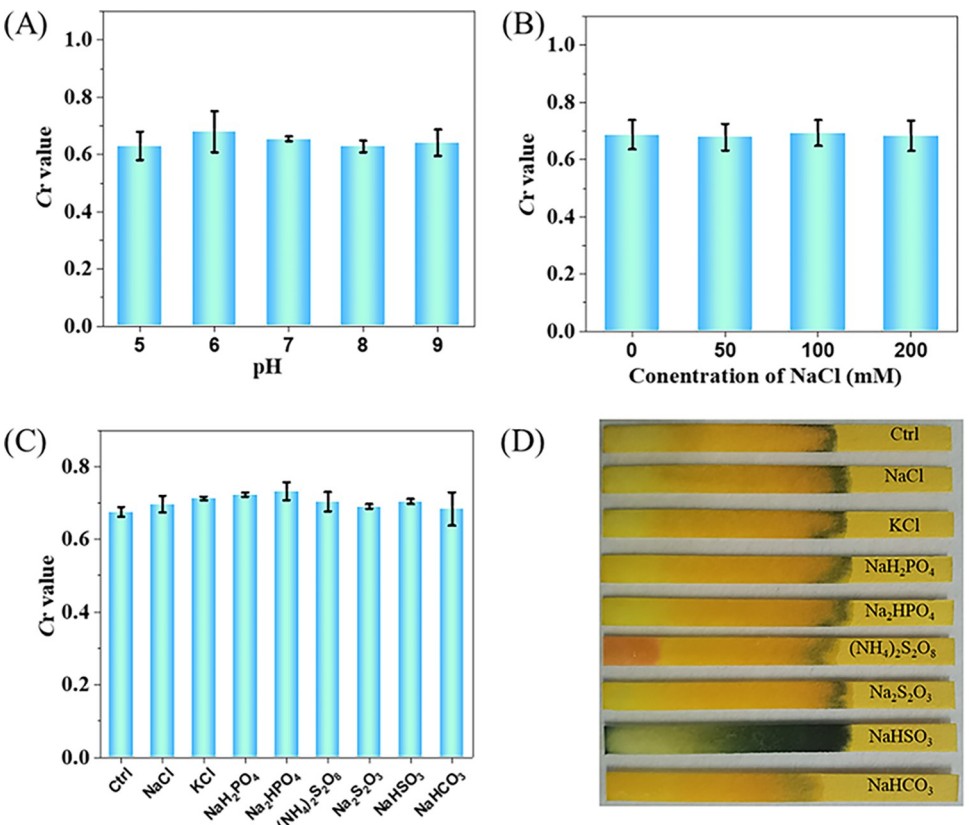

**Fig 4. The selectivity of the paper sensor.** (A) The $Cr$ values responses for the mixture solution of Alg and $Ca^{2+}$ with different pH values, (B) for aqueous solutions with NaCl solutions (0 to 200 mmol $L^{-1}$) and $Ca^{2+}$, (C) for aqueous solutions with other ions and $Ca^{2+}$, and (D) the corresponding photohraphs, respectively. The concentration of each ion was 10 mmol $L^{-1}$.

### The semi-quantitative detection of water hardness using the paper sensor

The World Health Organization recommends that the $Ca^{2+}$ level in drinkable water should be less than 5 mmol $L^{-1}$. Thus, the corresponding $Cr$ value of drinkable water sample detected by the paper sensor should be in the range of 0.259 to 0.595 in the absence of $Mg^{2+}$ according to Fig 3B. If only $Mg^{2+}$ is existing in the tested sample, the concentration of $Mg^{2+}$ should be below than 12.5 mmol $L^{-1}$ $Mg^{2+}$ for drinkable water, which means the same water hardness and calculated by the equation:

$$\text{Water hardness} = Ca^{2+}(g\ L^{-1}) \times 2.5 + Mg^{2+}(g\ L^{-1}) \times 4.1.$$

According to Fig 5, the $Cr$ values should be below 0.419. In short, the water hardness is suitable for drinking if the $Cr$ values lie in the range of 0.259 to 0.419, and unqualified with $Cr$ value above 0.595. Thus, the proposed sensor has been successfully applied for the semi-quantitative detection of water hardness.

### Comparison of the paper sensor with the commercial kit

The applicability of the paper sensor was also evaluated in water. The standard addition method was used herein. After 4 mmol $L^{-1}$ of $Ca^{2+}$ was initially added into the tested purified water, the additional different concentrations of $Ca^{2+}$ with 0, 2, 4 and 6 mmol $L^{-1}$ were spiked

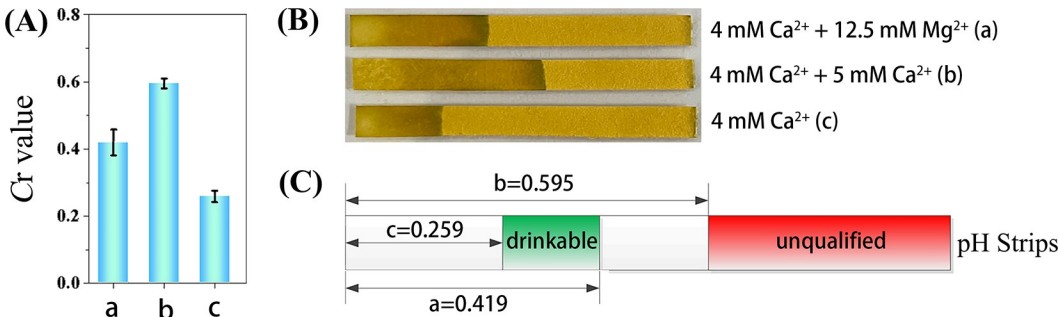

**Fig 5. The illustration of the semi-quantitative detection of water hardness using the paper sensor.** (A) The $Cr$ values and (B) corresponding images of the **paper** sensor under different conditions: in the presence of 4 mmol $L^{-1}$ $Ca^{2+}$ + 12.5 mmol $L^{-1}$ $Mg^{2+}$, 4 mmol $L^{-1}$ $Ca^{2+}$ + 5 mmol $L^{-1}$ $Ca^{2+}$, and 4mmol $L^{-1}$ $Ca^{2+}$. (C) The $Cr$ values ranges illustration of the sensor for the water hardness detection for the drinking water.

respectively. The concentrations of $Ca^{2+}$ were determined by the paper sensor (S5 Fig), the water hardness kit and inductively coupled plasma-mass spectrometry (ICP-MS), respectively. The recoveries were found varied from 95% to 108.9% and from 105% to 145% for the paper-based sensor and the commercial kit, respectively (Table 1). The results also exhibited the comparable accuracy of the proposed paper sensor with the commercial kit. Test-t analysis was performed between the proposed paper-based method and the ICP-MS method which is the gold standard for metal content measurement. The p values were calculated as 0.317, 0.353, and 0.596 respectively, corresponding to the marked samples with $Ca^{2+}$ concentrations of 2, 4, 6 mM. No significant difference were observed, which demonstrated the reliability of the paper method. More importantly, only 30 μL of the test sample is required for the paper sensor, which is much less than the amount of the test sample (10 mL) used for the commercial kit. In addition, the comparison of this paper-based sensor with different methods for the detection of $Ca^{2+}$ and $Mg^{2+}$ were performed (S1 and S2 Tables), which clearly demonstrated the merits of low-cost, convince, and easily-read for the paper-based sensor. Thus, this method works as an effective means for the quantification of $Ca^{2+}$ in water.

## Conclusions

In summary, this work reported a portable distance-based lateral flow paper-based sensor for drinking water hardness semi-quantitative detection based on the phase separation induced by

**Table 1. Hardness detection in purified water by the paper-based sensor, commercial kit and ICP-MS.**

| Method | Additional added $Ca^{2+}$ (mM) | Found (mM) | Recovery (%) | RSD (%) |
|---|---|---|---|---|
| Paper-based sensor | 2.0 | 2.1 | 108.9 | 4.4 |
| | 4.0 | 3.8 | 95.0 | 7.3 |
| | 6.0 | 5.7 | 95.0 | 7.4 |
| Water hardness kit | 2.0 | 2.9 | 145.0 | 12.9 |
| | 4.0 | 5.0 | 125.0 | 7.5 |
| | 6.0 | 6.3 | 105.0 | 7.8 |
| ICP-MS | 2.0 | 2.0 | 100.0 | 1.5 |
| | 4.0 | 4.1 | 102.5 | 8.8 |
| | 6.0 | 5.9 | 98.3 | 8.9 |

mM, mmol/L; RSD, relative standard deviation; ICP-MS, inductively coupled plasma-mass spectrometry.

the gel-sol transition. Briefly, the chelation between sodium alginate and $Ca^{2+}/Mg^{2+}$ triggers the formation of hydrogel to cause the viscosity changes of sodium alginate solution. The viscosity change can be quantified by the water flow distance on test strips. This method effectively avoids the usage of a large amount of samples, complicated operation, and high cost, which also exhibits high accuracy performance. Though the paper sensor could not exclude thoroughly the interference produced by other metal ions in some specific water samples, this approach can eliminate the error caused by the subjective color endpoint judgment in tested commercial kit, which shows the great potential of the paper sensor for further application.

## Supporting information

**S1 Fig. The photographs of the paper-based sensors in response to different concentrations of $Mg^{2+}$ in the presence of 0.2 wt% Alg.**
(TIF)

**S2 Fig. The condition optimization of the sensor.** (A) The performance of the sensor towards different concentrations of sodium alginate alone (0.1 wt%, 0.2 wt%, 0.3 wt%, 0.4 wt% and 0.5 wt%), (B-F) the images of the sensors towards the different concentrations of sodium alginate with different concentrations $Ca^{2+}$ (0, 2, 4, 10 and 20 mmol $L^{-1}$).
(TIF)

**S3 Fig. PThe images of the paper-based sensor in response to different concentrations of $Ca^{2+}$.**
(TIF)

**S4 Fig. The selectivity evaluation of the paper-based sensor.** (A) The images of the paper-based sensor towards the aqueous solution with different pH value; (B) The images of the paper-based sensor with the coexistence of different concentrations NaCl (0, 50, 100 and 200 mmol $L^{-1}$) with $Ca^{2+}$ (10 mmol $L^{-1}$), respectively.
(TIF)

**S5 Fig. The images of the paper-based sensors in response to different spiked concentration of $Ca^{2+}$.**
(TIF)

**S1 Table. Comparison of this work with different methods for the detection of $Ca^{2+}$.**
(DOCX)

**S2 Table. Comparison of this work with different methods for the detection of $Mg^{2+}$.**
(DOCX)

## Acknowledgments

We thank Yanhui Bi and Binglu Zhao for expert technical assistance.

## Author Contributions

**Conceptualization:** Qiongzheng Hu.

**Formal analysis:** Yulin Liu, Longzhan Dong, Wenli Wu, Jiantao Ping, Jingbo Chen, Qiongzheng Hu.

**Funding acquisition:** Yulin Liu, Wenli Wu, Jiantao Ping, Qiongzheng Hu.

**Investigation:** Yulin Liu, Longzhan Dong.

**Methodology:** Qiongzheng Hu.

**Project administration:** Qiongzheng Hu.

**Supervision:** Jingbo Chen, Qiongzheng Hu.

**Visualization:** Yulin Liu.

**Writing – original draft:** Yulin Liu, Jingbo Chen.

**Writing – review & editing:** Wenli Wu, Jiantao Ping, Qiongzheng Hu.

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
