## [Decision Letter · Decision Letter 0]

27 Feb 2024

PONE-D-24-04161A portable lateral flow distance-based paper sensor for drinking water hardness testPLOS ONE

Dear Dr. liu,

Thank you for submitting your manuscript to PLOS ONE. After careful consideration, we feel that it has merit but does not fully meet PLOS ONE’s publication criteria as it currently stands. Therefore, we invite you to submit a revised version of the manuscript that addresses the points raised during the review process.

We look forward to receiving your revised manuscript.

Kind regards,

Elingarami Sauli, PhD

Academic Editor

PLOS ONE

Journal Requirements:

"Natural Science Foundation of Shandong Province (ZR2021QH106, ZR2020QB153, ZR2022YQ123); the Taishan Scholars Program (tsqn201812088); the Education and Industry Integration Pilot Project of Qilu University of Technology (2023PY058, 2022PY036)"

Please state what role the funders took in the study.  If the funders had no role, please state: ""The funders had no role in study design, data collection and analysis, decision to publish, or preparation of the manuscript."" If this statement is not correct you must amend it as needed. 

"Acknowledgements This work was supported by the Natural Science Foundation of

Shandong Province (ZR2021QH106, ZR2020QB153, ZR2022YQ123); the Taishan

Scholars Program (tsqn201812088); the Education and Industry Integration Pilot

Project of Qilu University of Technology (2023PY058, 2022PY036)."

Please be informed that funding information should not appear in the Acknowledgments section or other areas of your manuscript. We will only publish funding information present in the Funding Statement section of the online submission form. 

"Natural Science Foundation of Shandong Province (ZR2021QH106, ZR2020QB153, ZR2022YQ123); the Taishan Scholars Program (tsqn201812088); the Education and Industry Integration Pilot Project of Qilu University of Technology (2023PY058, 2022PY036)"

6. Please include a copy of Table 1 which you refer to in your text on page 17 (in PDF format). 

**Additional Editor Comments:**

When responding to reviewer comments please make sure to improve the abstract, introduction, and discussion  parts, not forgetting to clearly indicate the innovation in your work.

Reviewers' comments:

Reviewer's Responses to Questions

**Comments to the Author**

1. Is the manuscript technically sound, and do the data support the conclusions?

Reviewer #1: Partly

Reviewer #2: Yes

Reviewer #3: Yes

2. Has the statistical analysis been performed appropriately and rigorously? 

Reviewer #1: N/A

Reviewer #2: Yes

Reviewer #3: Yes

3. Have the authors made all data underlying the findings in their manuscript fully available?

Reviewer #1: Yes

Reviewer #2: Yes

Reviewer #3: Yes

4. Is the manuscript presented in an intelligible fashion and written in standard English?

Reviewer #1: No

Reviewer #2: Yes

Reviewer #3: Yes

5. Review Comments to the Author

Reviewer #1: This manuscript reported a portable lateral flow distance-based paper sensor for drinking water hardness test. However, the manuscript is not well-written. Here gives some suggestions for improvement of the manuscript:

1. The abstract section needs to be revised, more information should be mentioned, such as the recoveries of the tested purified water.

2. The third paragraph of Introduction needs to be revised. The paper-based lateral flow sensor and its applications should be introduced first, followed by the distance signal readout method and its applications, and finally the analyte-responsive hydrogels.

3. “Condition optimization of the paper sensor” section, the author declared that “Moreover, the similar variation tendency was found for the solution and the mixture of Alg and Ca2+, respectively.” The data should be provided.

4. The author declared that “The concentrations of Ca2+ were determined by the paper sensor, the water hardness kit and inductively coupled plasma-mass spectrometry (ICP-MS) (Fig.S5).” However, Fig.S5 is just a graph of the paper-based sensor in response to different added concentration of Ca2+.

5. Table 1 is missing.

6. In Fig. S2 captions, the concentrations of Ca2+ (0, 1, 2, 5 and 10 mM) is wrong.

7. The discussion of Fig.S4 in the main text is missing. Moreover, other metal ions in the water sample can also affect the sensor's results, not just chloride ions.

Reviewer #2: In the present manuscript, the authors have made an interesting design of A portable lateral flow distance-based paper sensor for testing hardness of drinking water. The study is simple therefore, novelty should be highlighted. A thorough literature survey must be included. The characterizations especially for the paper based sensors should be improved. The study related to repeatability and stability must be presented. Paper can be accepted after these minor revision.

Reviewer #3: The authors developed a simple and portable lateral flow sensor with distance reading signal for the semiquantitative detection of drinking water hardness. Where the Ca2+/Mg2+ concentration can be determined by the water lateral flow distance on paper strips of the upper aqueous solution. Several studies have been carried out, including 1) Optimization of sodium alginate concentration, 2) Determination of water hardness using commercial test kit, 3) Feasibility of Ca2+/Mg2+ detection of the paper sensor, 4) Condition optimization of the paper sensor, 5) Detection of Ca2+ and Mg2+, 6) The semi-quantitative detection of water hardness using the paper sensor and 7) Comparison of the paper sensor with commercial kit for Ca2+ determination. Sample analyze was also performed. The article shows important results for the area of knowledge. However, the manuscript presents few points that should be improved to final decision.

Required comments:

1) Review the units. Preferably use mol L-1, g L-1, Temperature in K, etc. According to the international system of units.

2) Does study involving pH and temperature variation is necessary for the optimization of sodium alginate concentration?

3) Did the authors measure sensor stability?

4) If possible, include a table comparing the values obtained with literature data from other sensors.

6. PLOS authors have the option to publish the peer review history of their article (what does this mean?). If published, this will include your full peer review and any attached files.

Reviewer #1: No

Reviewer #2: **Yes: **Dr Khairunnisa Amreen

Reviewer #3: **Yes: **Emerson Schwingel Ribeiro

---

## [Author Response · Author response to Decision Letter 0]

11 Apr 2024

We would first like to appreciate the editors and the reviewers for your positive and constructive comments and suggestions, and give us a chance to improve our manuscript.

Responses to Editor:

Question 1: Please ensure that your manuscript meets PLOS ONE's style requirements, including those for file naming.

Answer: Thanks for your careful comments. We have modified our manuscript as PLOS ONE's style requirements. 

Question 2: We note that the grant information you provided in the‘Funding Information’ and ‘Financial Disclosure’ sections do not match.

Answer: Thanks for your friendly suggestion. We have provided the correct grant numbers for the awards for our study in the ‘Funding Information’ section. 

Question 3: Please state what role the funders took in the study.

Answer: Thanks a lot for your kind suggestion. The funders had no role in study design, data collection and analysis, decision to publish, or preparation of the manuscript. We have included Role of Funder statement in our cover letter.

Question 4: Please remove any funding-related text from the manuscript and let us know how you would like to update your Funding Statement.

Answer: Thank you for your kind reminder. We have removed any funding-related text from the manuscript. We will no longer update our Funding Statement.

Question 5: Please ensure that you have an ORCID iD.

Answer: Thanks a lot for your kind suggestion. I have an ORCID iD which is 0000-0003-1801-9351.

Question 6: Please include a copy of Table 1 which you refer to in your text on page 17 (in PDF format).

Answer: Thanks for your careful and kind suggestion. We apologize for our carelessness. We have added Table 1 to the latest version.

Question 7: Please include captions for your Supporting Information files at the end of your manuscript, and update any in-text citations to match accordingly

Answer: Thanks for your critical comments. We have attached the captions of Supporting Information files at the end of our manuscript, updated any in-text citations to match accordingly.

Question 8: Please review your reference list to ensure that it is complete and correct.

Answer: Thanks a lot for your kind suggestion. We have carefully reviewed our references list. We ensure that it is complete and correct.

Responses to Reviewer #1:

Question 1: The abstract section needs to be revised, more information should be mentioned, such as the recoveries of the tested purified water.

Answer: Thanks for your professional comments. We have added the recoveries of paper-based sensor into abstract section.

Question 2: The third paragraph of Introduction needs to be revised.

Answer: Thanks for your professional suggestion which will significantly improve the logic of our article. We have adjusted the order of introduction for paper-based lateral flow sensor , distance signal readout method and analyte-responsive hydrogels.

Question 3: “Condition optimization of the paper sensor” section, the author declared that “Moreover, the similar variation tendency was found for the solution and the mixture of Alg and Ca2+, respectively.” The data should be provided.

Answer: Thank you for your critical comments. We apologize for our oversight. Originally, we wanted to declared that “the similar variation tendency was found for the solution and the mixture of Alg and Mg2+”, as shown in Fig S1.

Question 4: The author declared that “The concentrations of Ca2+ were determined by the paper sensor, the water hardness kit and inductively coupled plasma-mass spectrometry (ICP-MS) (Fig.S5).” However, Fig.S5 is just a graph of the paper-based sensor in response to different added concentration of Ca2+.

Answer: Thanks for your careful and kind suggestion. We apologize for our carelessness. Fig S5 is a graph of the paper-based sensor in response to different added concentration of Ca2+. We have modified it to “The concentrations of Ca2+ were determined by the paper sensor (Fig S5), the water hardness kit and inductively coupled plasma-mass spectrometry (ICP-MS).”

Question 5: Table 1 is missing.

Answer: Thanks a lot for your careful comment. We have added Table 1 to the latest version.

Question 6: In Fig. S2 captions, the concentrations of Ca2+ (0, 1, 2, 5 and 10 mM) is wrong.

Answer: Thanks for your careful review. We apologize for our oversight. We have modified the concentration in Supporting Information.

Question 7: The discussion of Fig.S4 in the main text is missing. Moreover, other metal ions in the water sample can also affect the sensor's results, not just chloride ions.

Answer: Thanks for your professional review. In the main text, we added a brief discussion about the anti-interference capacity of the paper sensor for Ca2+ detection, as shown in Figs 1-4 and Fig S4. 

Responses to Reviewer #2:

Question 1:A thorough literature survey must be included. The characterizations especially for the paper based sensors should be improved. The study related to repeatability and stability must be presented.

Answer: Thanks for your professional suggestions. 

We made comparison of this work with different methods for the detection of Ca2+ and Mg2+ by a thorough literature survey, which were presented as Tables S1 and S2 in Supporting Information.

In addition, we apologize for missing Table 1 in uploaded manuscript. In Table 1, the RSD of Paper-based sensor ranges from 4.4 to 7.4, which indicates good stability of this method. We have added Table 1 to the latest version.

Responses to Reviewer #3:

Question 1:Review the units. Preferably use mol L-1, g L-1, Temperature in K, etc. According to the international system of units.

Answer: Thanks a lot for your kind suggestion. We have reviewed all the units in the manuscript. According to the international system of units, we have converted the units in the manuscript.

Question 2: Does study involving pH and temperature variation is necessary for the optimization of sodium alginate concentration?

Answer: Thanks for your valuable suggestion. We have tested the anti-interference capacity of the paper sensor for Ca2+ detection including pH and ionic strength. The ionic strength in actual drinking water samples exhibits simple, which rarely contains heavy metal cations. Therefore, we selected common anions for anti-interference detection. 

The effect of different temperatures on the formation of calcium oleate has been observed in our previous study. With the increase in temperature, the amount of calcium oleate particles on the wall of the centrifugal tube decreased signiffcantly due to the increase in its solubility. However, the reaction time was largely prolonged if the temperature was too low. Therefore, the temperature of calcium oleate formation was selected as 298K[1].

The corresponding results were represented in Results and discussion.

Reference

[1] Xia S, Yin F, Xu L, Zhao B, Wu W, Ma Y, et al. Paper-Based Distance Sensor for the Detection of Lipase via a Phase Separation-Induced Viscosity Change. Anal Chem . 2022; 94(49): 17055-17062. 

Question 3: Did the authors measure sensor stability? 

Answer: Thanks for your valuable suggestion. We apologize for missing Table 1 in uploaded manuscript. In Table 1, the RSD of Paper-based sensor ranges from 4.4 to 7.4, which indicates good stability of this method. We have added Table 1 to the latest version.

Question 4: If possible, include a table comparing the values obtained with literature data from other sensors.

Answer: Thanks a lot for your professional comment. We added Tables S1 and S2 for the comparison of existing sensor, which were attached to Supporting Information.

---

## [Decision Letter · Decision Letter 1]

22 May 2024

PONE-D-24-04161R1A portable lateral flow distance-based paper sensor for drinking water hardness testPLOS ONE

Dear Dr. liu,

Thank you for submitting your manuscript to PLOS ONE. After careful consideration, we feel that it has merit but does not fully meet PLOS ONE’s publication criteria as it currently stands. Therefore, we invite you to submit a revised version of the manuscript that addresses the points raised during the review process.

The authors should improve the introduction and discussion parts, including justification of the procedure used when compared with other similar methods.

We look forward to receiving your revised manuscript.

Kind regards,

Elingarami Sauli, PhD

Academic Editor

PLOS ONE

Journal Requirements:

Reviewers' comments:

Reviewer's Responses to Questions

**Comments to the Author**

1. If the authors have adequately addressed your comments raised in a previous round of review and you feel that this manuscript is now acceptable for publication, you may indicate that here to bypass the “Comments to the Author” section, enter your conflict of interest statement in the “Confidential to Editor” section, and submit your "Accept" recommendation.

Reviewer #1: (No Response)

Reviewer #3: All comments have been addressed

Reviewer #4: (No Response)

Reviewer #5: (No Response)

2. Is the manuscript technically sound, and do the data support the conclusions?

Reviewer #1: Yes

Reviewer #3: Yes

Reviewer #4: Yes

Reviewer #5: No

3. Has the statistical analysis been performed appropriately and rigorously? 

Reviewer #1: Yes

Reviewer #3: Yes

Reviewer #4: (No Response)

Reviewer #5: No

4. Have the authors made all data underlying the findings in their manuscript fully available?

Reviewer #1: Yes

Reviewer #3: Yes

Reviewer #4: (No Response)

Reviewer #5: Yes

5. Is the manuscript presented in an intelligible fashion and written in standard English?

Reviewer #1: Yes

Reviewer #3: Yes

Reviewer #4: (No Response)

Reviewer #5: No

6. Review Comments to the Author

Reviewer #1: After the revisions, the paper improved significantly. The manuscript's structure, flow, or writing is acceptable for publication.

Reviewer #3: The authors made the necessary corrections presented by the reviewers. The article is ready to be published in Plos One.

Reviewer #4: This manuscript has been carefully revised according to the suggestions of first-round review. Followings are some suggestions for further revisions.

1, There are commercially available instrument for measuring the hardness of water, which like the pH meter. Please also introduce them in the introduction section, and include this type instrument for comparison in the measurement of hardness of water in the experimental section.

2, According to the Reviewer #1 of first-round review, there are several errors in the manuscript. Please double check the manuscript.

3, Do not use non-English terms such as "anti-interference" or "foreign ions". Make use of correct terms "Selectivity" and "potentially interfering ions", respectively.

4, Please avoid using first-person narratives for a scientific paper.

5, The shortcomings of present work can be emphasized in the conclusion section.

6, Please carefully check and unify the units. For example, both the “mM” and “mmol-1” have been used in the manuscript.

Reviewer #5: The manuscript "A portable lateral flow distance-based paper sensor for drinking water hardness test" describes a paper-based device for the determination of water hardness, expressed as Ca2+ and Mg2+ concentration, based on the formation of a hydrogel after the chelation with sodium alginate. Although the idea of the study is interesting, I do not think the quality of the manuscript support its publication. To be considered, an improvment must be performed. Some specific comments:

a) The English language throughout the manuscript must be revised. There are many typos (e.g. thrice) and strange sentences (e.g. Drinking water quality is a problem concerned by people). The peer-reviewing is not a proof reading service, so the English must be revised by someone dedicated to it.

b) A broader literature revision must be done. There are, at least, two papers dealing with the determination of water hardness using paper-based devices that were not mentioned (10.1016/j.measurement.2020.108085 and 10.3390/su14063363). It should be inserted and discussed.

c) Eriochrome T is the most usual "reagent" for the determination of water hardness. Why your procedure is better?

d) "anti-interference" sounds strange

e) The "Cr values" is not clear how to calculate. Please clarify it.

f) The images were taken with no illumination control? Is the results affected by different illumination conditions? It should be discussed.

g) Test-t should be performed to compare the results obtained by the proposed method with the reference ones.

7. PLOS authors have the option to publish the peer review history of their article (what does this mean?). If published, this will include your full peer review and any attached files.

Reviewer #1: No

Reviewer #3: **Yes: **Emerson Schwingel Ribeiro

Reviewer #4: No

Reviewer #5: No

---

## [Author Response · Author response to Decision Letter 1]

6 Jul 2024

We would like to appreciate the editors and the reviewers for the positive and constructive comments and suggestions, and give us a chance to improve our manuscript.

Responses to Reviewer #4:

Question 1: There are commercially available instrument for measuring the hardness of water, which like the pH meter. Please also introduce them in the introduction section, and include this type instrument for comparison in the measurement of hardness of water in the experimental section.

Answer: Thanks for your careful and kind suggestions. We investigate carefully all the available related instrument. Several methods for measuring the hardness of water have been reported. Among them, atomic absorption spectrometry and complexometric titration are the classic methods. Additionally, the fluorescence method and ionophore-based ion-selective optode also have been used for for monitoring water hardness. The above content has been included in the introduction section. In the study, we employed the commercial kit for Ca2+ determination in control group in the experimental section.

Question 2: According to the Reviewer #1 of first-round review, there are several errors in the manuscript. Please double check the manuscript.

Answer: Thanks for your kind suggestion. We have carefully corrected the errors pointed out by the Reviewer #1. Meanwhile, we checked the manuscript carefully. 

Question 3: Do not use non-English terms such as "anti-interference" or "foreign ions". Make use of correct terms "Selectivity" and "potentially interfering ions", respectively.

Answer: Thanks for your professional suggestion which will significantly improve the readability of our article. We have adjusted non-English terms according to your advice .

Question 4: Please avoid using first-person narratives for a scientific paper.

Answer: Thanks for your careful and kind suggestion. We have switched from first-person narratives to third-person narratives to present scientific facts.

Question 5: The shortcomings of present work can be emphasized in the conclusion section.

Answer: Thanks for your professional comments. We have emphasized the shortcomings of present work in the conclusion section.

Question 6: Please carefully check and unify the units. For example, both the “mM” and “mmol-1” have been used in the manuscript.

Answer: Thank you for your critical comments. We apologize for our oversight. We have carefully checked and unified the units as mmol L-1.

Responses to Reviewer #5

Question 1: The English language throughout the manuscript must be revised. There are many typos (e.g. thrice) and strange sentences (e.g. Drinking water quality is a problem concerned by people). The peer-reviewing is not a proof reading service, so the English must be revised by someone dedicated to it.

 Answer: Thanks for your kind suggestion. We have carefully revised the English language throughout the manuscript. 

Question 2: A broader literature revision must be done. There are, at least, two papers dealing with the determination of water hardness using paper-based devices that were not mentioned (10.1016/j.measurement.2020.108085 and 10.3390/su14063363). It should be inserted and discussed.

Answer: Thank you for your valuable suggestion. We have unified and updated the format of the literature throughly. Meanwhile, we have added the mentioned literatures (10.1016/j.measurement.2020.108085 and 10.3390/su14063363) with discussion in the manuscript. 

Question 3: Eriochrome T is the most usual "reagent" for the determination of water hardness. Why your procedure is better?

Answer: Thanks for your professional comments. Eriochrome T is the most usual "reagent" for the determination of water hardness. In this study, the commercial kit for Ca2+ determination employed in the control group is made up of Eriochrome T. Both approaches have advantages and disadvantages. Our procedure provides a portable approach for semi-quantitative detection of drinking water hardness with convenience and low cost. However, the detection of water samples in complex environments needs to be further optimized.

Question 4: "anti-interference" sounds strange.

Answer: Thank you for your critical comments. We apologize for using non-English terms. We have replaced it with “Selectivity”.

Question 5: The "Cr values" is not clear how to calculate. Please clarify it.

Answer: Thanks a lot for your suggestions. The calculation of the coverage ratios of water mark was illustrated as below:

Pmark = the pixel values of the watermark area= 58765.

Ptotal= the pixel values of the whole test strip = 101797

Therefore, Cr = Pmark / Ptotal =58765/101797 = 0.577

Question 6: The images were taken with no illumination control? Is the results affected by different illumination conditions? It should be discussed.

Answer: Thanks for your professional comments. In this study, the photographs were taken at ambient light conditions. Moreover, the detection relies on measuring the water flow distance. It is not be affected by the illumination conditions. This is an advantage of the distance sensor.

Question 7: Test-t should be performed to compare the results obtained by the proposed method with the reference ones.

Answer: Thanks for your professional comments. We performed the test-t analysis between the proposed paper-based method and the ICP-MS method which is the gold standard for metal content measurement. The p values were calculated as 0.317, 0.353, and 0.596 respectively, corresponding to the marked samples with Ca2+ concentrations of 2, 4, and 6 mM, respectively. No significant difference was observed, which demonstrate the reliability of the paper-based method.

---

## [Decision Letter · Decision Letter 2]

24 Jul 2024

A portable lateral flow distance-based paper sensor for drinking water hardness test

PONE-D-24-04161R2

Dear Dr. Yulin Liu

We’re pleased to inform you that your manuscript has been judged scientifically suitable for publication and will be formally accepted for publication once it meets all outstanding technical requirements.

Kind regards,

Elingarami Sauli, PhD

Academic Editor

PLOS ONE

Additional Editor Comments (optional):

All reviewers' (original and secondary) comments have been addressed to my satisfaction. The submission can be accepted with minor editorial revisions as highlighted below.

Reviewers' comments:

Reviewer's Responses to Questions

**Comments to the Author**

1. If the authors have adequately addressed your comments raised in a previous round of review and you feel that this manuscript is now acceptable for publication, you may indicate that here to bypass the “Comments to the Author” section, enter your conflict of interest statement in the “Confidential to Editor” section, and submit your "Accept" recommendation.

Reviewer #5: All comments have been addressed

2. Is the manuscript technically sound, and do the data support the conclusions?

Reviewer #5: Yes

3. Has the statistical analysis been performed appropriately and rigorously? 

Reviewer #5: Yes

4. Have the authors made all data underlying the findings in their manuscript fully available?

Reviewer #5: Yes

5. Is the manuscript presented in an intelligible fashion and written in standard English?

Reviewer #5: No

6. Review Comments to the Author

Reviewer #5: The authors made susbtantial changes in the revised version of the manuscript. The results are now better presented and the introduction section has improved. Prior to the publication, some revisions are still needed.

a) Some expressions and sentences still needs improvment, such as: the usage of the word "Remarkably" in the abstract is unecessary. The sentence "Upon the Alg solution was incubation..." and "S2 Fig" need revision.

b) Only cations Na+ and K+ were evaluated as interferent. Why? More cations should be investigated.

7. PLOS authors have the option to publish the peer review history of their article (what does this mean?). If published, this will include your full peer review and any attached files.

Reviewer #5: No

---

## [Editor Report · Acceptance letter]

29 Jul 2024

PONE-D-24-04161R2 

PLOS ONE

Dear Dr. Liu, 

I'm pleased to inform you that your manuscript has been deemed suitable for publication in PLOS ONE. Congratulations! Your manuscript is now being handed over to our production team.

Kind regards, 

on behalf of

Dr. Elingarami Sauli 

Academic Editor

PLOS ONE